# Dysmenorrhoea: Can Medicinal Cannabis Bring New Hope for a Collective Group of Women Suffering in Pain, Globally?

**DOI:** 10.3390/ijms232416201

**Published:** 2022-12-19

**Authors:** Amelia Seifalian, Julian Kenyon, Vik Khullar

**Affiliations:** 1Department of Urogynaecology, St. Mary’s Hospital, Imperial College London, London W2 1NY, UK; 2The Dove Clinic for Integrated Medicine, Winchester SO21 1RG, UK

**Keywords:** medical cannabis, cannabinoids, tetrahydrocannabinol, dysmenorrhoea, pain, cancer

## Abstract

Dysmenorrhoea effects up to 90% of women of reproductive age, with medical management options including over-the-counter analgesia or hormonal contraception. There has been a recent surge in medicinal cannabis research and its analgesic properties. This paper aims to critically investigate the current research of medicinal cannabis for pain relief and to discuss its potential application to treat dysmenorrhoea. Relevant keywords, including medicinal cannabis, pain, cannabinoids, tetrahydrocannabinol, dysmenorrhoea, and clinical trial, have been searched in the PubMed, EMBASE, MEDLINE, Google Scholar, Cochrane Library (Wiley) databases and a clinical trial website (clinicaltrials.gov). To identify the relevant studies for this paper, 84 papers were reviewed and 20 were discarded as irrelevant. This review critically evaluated cannabis-based medicines and their mechanism and properties in relation to pain relief. It also tabulated all clinical trials carried out investigating medicinal cannabis for pain relief and highlighted the side effects. In addition, the safety and toxicology of medicinal cannabis and barriers to use are highlighted. Two-thirds of the clinical trials summarised confirmed positive analgesic outcomes, with major side effects reported as nausea, drowsiness, and dry mouth. In conclusion, medicinal cannabis has promising applications in the management of dysmenorrhoea. The global medical cannabis market size was valued at USD 11.0 billion in 2021 and is expected to expand at a compound annual growth rate (CAGR) of 21.06% from 2022 to 2030. This will encourage academic as well as the pharmaceutical and medical device industries to study the application of medical cannabis in unmet clinical disorders.

## 1. Introduction

Dysmenorrhoea is a medical term used to describe painful menstruation. Dysmenorrhoea is extremely common amongst women of reproductive age and is estimated to affect up to 90% of these women [1]. Primary dysmenorrhoea occurs as a result of increased prostaglandin release, amongst other chemical imbalances [2]. The chemical imbalance causes abnormal uterine contractions, interrupting blood flow and increasing anaerobic metabolites that, in turn, stimulate pain receptors. Current treatment options for primary dysmenorrhoea include analgesia or hormonal contraceptive methods. Secondary dysmenorrhoea occurs as a result of underlying gynaecological disease, such as endometriosis, which is managed by treating the underlying cause. All further mentions of dysmenorrhoea in this paper refer to primary dysmenorrhoea [3,4].

A new proposed strategy entering the commercial market for the management of severe dysmenorrhoea is the application of cannabidiol (CBD)-based products over the counter (OTC). Naturally, this leads to potential application of potent cannabis-based products for medicinal use (CBPMs) to treat dysmenorrhoea. CBPMs are defined as any product containing cannabis or its resin, cannabinol, or a cannabinol derivative for medicinal use in humans [3]. The most common forms of CBPMs include flower for inhalation or oil for ingestion. Tetrahydrocannabinol (THC) is a chemical compound found in the cannabis plant known for its psychoactive properties, hence recognised as a drug of abuse for recreational use. CBD is also found in the cannabis plant. CBD does not have psychoactive properties; however, it remains regulated in a majority of countries due to its associations with THC [4].

There has been a recent surge in the availability of CBD products, with extensive commercialisation of its holistic and cosmetic applications. However, the CBD products available OTC contain very small amounts of CBD and are unlikely to result in any meaningful positive effect [5]. CBD is not considered to be a CBPM since it is an isolate that does not contain any THC. Therefore, OTC CBD is exempt from cannabis-related laws in the majority of countries. It is important to note, however, that CBD is known to be difficult to isolate, with trace THC content detectable in a number of CBD products. This causes complications surrounding the legalities in countries where CBPM is regulated but CBD is exempt. The complexities are extensive and vary from country to country, thus creating obstacles for patient access.

Recent years have seen an overall movement towards legalisation and freedom of access of CBPMs. Cannabis has been legalised in a number of countries, including 33 states in America, and decriminalised for possession in several more. There has been steady progress towards availability and in the UK, drug scheduling changed in 2018 [6], allowing specialist prescriptions of CBPM. However, changes in laws have not always been reflected in practice, with only 328 prescriptions of CBPMs in the UK by early 2020 from the changes in 2018 [7].

Current evidence-based applications of CBPMs broadly include chronic pain, chemotherapy-induced nausea and vomiting, multiple sclerosis-related spasticity, and epilepsy [8]. Cannabis is the most commonly used illegal drug worldwide, yet it is still often restricted for healthcare prescribers and heavily stigmatised amongst healthcare staff [9,10]. The aim of this research was to critically evaluate CBPMs for their analgesic application in the management of dysmenorrhoea.

## 2. Dysmenorrhoea—Its Clinical Impact and Current Management

Dysmenorrhoea is not seen as a medical condition, but rather a pain that the majority of women are expected to bear. Yet the severity of this pain can be debilitating, causing work-related absenteeism, and impacting activities of daily life. There is a high volume of research linking chronic pain to depression, anxiety, and smoking habits [11,12]. There have been multiple studies linking the negative impact of dysmenorrhoea to stress and poor mental health outcomes [13]. Risk factors for dysmenorrhoea include smoking, nulliparity, heavy menstrual periods, young age—less than 20 years old—and a background of mental health issues [14].

Primary dysmenorrhoea occurs as a result of hypercontractility of the uterus—hence the main aim of treatment is to support uterine relaxation whether using drugs or non-medicinal techniques. The severity of pain experienced in dysmenorrhoea has been linked to the amount of excess prostaglandin released [15]. The pain described during dysmenorrhoea is typically lower abdominal, radiating to the back or legs, cramping in character, lasting eight to 72 h and starting at the onset of menstruation [16]. The pain therefore has an expected onset in patients with regular periods, is constant and lasts for a certain number of days. These are important to consider when planning appropriate analgesia to manage and suppress pain.

First line pharmacological treatment of dysmenorrhoea is with non-steroidal anti-inflammatory drugs (NSAIDs), which have been proven to be effective [17] (Paracetamol, considered less effective than Ibuprofen [18,19,20]), or oral contraceptives (OC) to prevent menstruation and the direct cause of pain in the first instance. These are the mainstay of drugs used in clinical practice with other medicines and non-medical therapies being investigated by research teams, but none yet found to be more effective than current practice. The alternative pathways being investigated for analgesic properties include use of medicinal plants [21], exercise [22] and acupuncture [23], amongst other homeopathic pathways.

Long-term NSAID use has been linked to organ damage as well as several adverse effects. Adverse outcomes include gastrointestinal ulcers, which can lead to iron deficiency anaemia, amongst other syndromes; higher risk of cardiovascular events, including myocardial infarction; and kidney damage, such as acute kidney injury or chronic kidney disease [24,25]. OC act on hormonal pathways and are unsuitable to prescribe where the patient cannot tolerate the side effects such as nausea, headaches, and weight gain [26]. Additionally, OC are unsuitable for a population of patients planning to conceive and are often stigmatised in certain religious and cultural settings.

## 3. Cannabis-Based Medicines and Their Properties

There are over 100 phytocannabinoids found in the cannabis plant, with the most abundant being THC, followed closely by CBD. There are several chemovars of cannabis, which are divided according to their chemical composition. Cannabis can be split into a number of categories, often divided to marijuana or hemp types, containing various ratios of THC versus CBD. A THC concentration of less than 0.3% is required to be classified as hemp [27]. In this paper, references to cannabis or cannabis plants include only the marijuana type. For best clinical practice, it is important to consider the clinical properties of the chemovar being prescribed prior to dispensing [28].

THC is a lipid found in all aerial parts of the cannabis plant (Figure 1) but most abundant in the flowers of the female plant, particularly the cannabis sativa species. At room temperature, THC is a glassy solid with the molecular formula C_21_H_30_O_2_ [29]. THC is soluble in lipids and solvents but has weak solubility in water [30]. THC is psychoactive and reacts with endocannabinoid (EC) receptors found in the brain, causing mood changes, euphoria and the commonly known “high” feeling [31].

CBD is a chemical compound found naturally in the resin of the cannabis plant. The molecular formula of CBD is the same as that of THC, C_21_H_30_O_2_ [32]. The difference between CBD and THC is a marginal distinction in the bonding structure—CBD has a hydroxyl group in place whereas THC has a cyclic ring, as seen in Figure 2. This difference in bonding structure has a great impact on its molecular interaction with EC receptors and thus the medicinal properties.

CBD is a hydrophobic compound needing emulsification to exist in aqueous solution form and is soluble in lipid and organic solvents, including alcohols [33]. CBD is a clear crystalline solid at room temperature with a melting point of 62–63 °C [34]. Both pyrolysis and certain acidic conditions may convert CBD to THC, and therefore, if pure CBD is sought, all processes that may heat the compound must be avoided [35].

CBPMs are prescribed as oil, for oral ingestion, or as flower buds, for smoking or vaping. The method of administration of CBPMs, oral versus inhaled, can impact clinical outcomes—including the onset of action and duration of action of the drug. See Table 1 for an appropriate summary of oral versus inhaled administration of CBPMs [36,37]. Oral preparations have delayed onset but act over a longer period compared to inhaled preparations, which act much faster but in conjunction dissipate faster, hence resulting in shorter-term effects. An appropriate prescription for primary dysmenorrhoea would be for daily oral ingestion to manage symptoms with inhalation for breakthrough pain.

There are two methods of extracting and isolating phytochemicals from the cannabis plant: the carbon dioxide (CO_2_) extraction method and the solvent extraction method. The carbon dioxide (CO_2_) extraction method involves a pressurised chamber compressing CO_2_ gas into liquid form, which is then forced over the cannabis plant, thus extracting the CBD and THC. This method is the most safe and effective—producing the purest form of CBPM [38]. However, the CO_2_ extraction method is the most expensive, with the machinery costing between GBP 95,000–GBP 110,000 [39]. The solvent extraction method involves soaking the plant in the chosen liquid solvent, often food-grade ethanol, or a hydrocarbon—such as propane or butane. The remaining solution is then either evaporated or distilled, leaving an oil resin of phytochemicals that include both THC and CBD. This method of extraction is more hazardous with use of flammable and toxic solvents and less efficient with weaker product outcomes. However, it is the cheapest and fastest method of extraction with minimal expenses and a simple process.

## 4. Application of Cannabis-Based Medicines Clinically

CBPMs have recently risen to media attention as their potential applications for medicinal use have increased. However, clinical specialists often lack clear guidance on how to safely prescribe CBPMs and surrounding laws, depending on each individual country of practice. CBPMs have been applied to manage pain, inflammation, cancer, epilepsy and seizures, addiction, and mental health (for example, anxiety and insomnia). Clinicians favour using higher concentrations of CBD compared to THC, due to evidence that higher CBD doses may negate the psychoactive effect of THC, although there is heterogeneity in results, and further confirmation is required as recent research has shown that this effect is not translated in clinical practice [40,41]. Cancer is a global leading cause of mortality and novel cancer treatments are continuously under research; high-dose CBDs are currently under investigation for their anticancer properties to treat a vast spectrum of cancers [42].

In the UK, CBPMs are entering the healthcare field with guideline support by the National Institute for Health and Care Excellence (NICE). The UK National Health Service (NHS) currently provides cannabis-based products for only three conditions; all conditions must be treatment-resistant with at least three appropriate options tried and failed.

In the UK, Epidiolex is prescribed for patients suffering from Dravet syndrome or Lennox–Gastaut syndrome—both of which are rare and severe forms of childhood epilepsy that can result in death. Epidiolex is an oral solution containing highly purified CBD liquid; it is used to treat children with more severe forms of epilepsy [43]. There are concerns regarding the long-term effects on cognition, development, and behaviour. However, initial studies are promising, indicating no adverse effects at one-year follow up [44]. Further studies on safety are needed to allow confident and safe prescription of Epidiolex.

Nabiximols, a naturally occurring cannabinoid extract, is an oral mouth spray containing both THC and CBD in roughly equal amounts. Nabiximols are prescribed to manage the muscle spasms and stiffness caused by multiple sclerosis [45]. The THC present can cause a euphoric “high”; however, side effects are reported to be mild to moderate [46].

## 5. Mechanisms of Cannabis-Based Medicines for Pain Relief

The EC system consists of a complex network of cell signalling pathways that modulate a number of responses in the human body, including inflammatory responses and pain signalling pathways. There are two types of EC G-protein coupled receptors, cannabinoid receptors 1 (CB1R) and cannabinoid receptors 2 (CB2R). CB1R can be found both in the central and peripheral nervous system, whereas CB2R is more abundant in the peripheral nervous system [47]. Interactions with central CB1R causes alterations in mood and euphoria [48]. CBPMs interact with the EC system of the human body, hence modulating the cell signalling pathways with potential analgesic effect.

CB2R receptors modulate the cardiovascular, gastrointestinal, metabolic, immune respiratory and reproductive systems amongst others, with focus on peripherally acting systems [49,50]. Acute pain and chronic inflammatory pain have different mechanisms; CB2R is thought to have an analgesic effect on chronic inflammatory pain rather than with acute pain [51]. THC acts as a partial agonist to both CB1R and CB2R [52]. CB1R is the most predominant EC receptor, highly dense in areas of the brain controlling behaviour, mood and memory [53]. CBD is a negative allosteric modulator of CB1R that weakens the ability of THC to bind to this receptor and so is considered to possibly dampen the psychoactive effects of THC, although this has not yet been confirmed by clinical research [40,54].

Transient receptor potential (TRP) channels are a group of ionotropic cation receptors located on several types of animal cell [55]. The TRP vanilloid (TRPV) subfamily of channels has significant involvement with pain signalling, and TRP dysfunction has been identified in neuropathic pain and inflammation, hence modulation of these channels may provide analgesic benefit [56]. Both CBD and THC are agonists of the TRPV channels, which may play a role in furthering their analgesic properties [57]. There are numerous more interactions of CBD/THC with enzymes and receptor signalling pathways in the human body, which may impact its properties of action for use with cannabinoid receptors and TRP being most significant.

CB1R, CB2R and TRPV have all been located in human normal myometrial tissue [58]. The uterus is the site of contractility, with the myometrium being the muscle that causes contractility. Research has shown that the EC system has a role in myometrial contractility during menstruation and that cannabinoid agonists, such as THC, are able to trigger myometrial relaxation—this would, in turn, ease the pain of dysmenorrhoea [59]. Although there is little research on the subject, it is thought that CB1R has a greater role in myometrium relaxation [60]. Dysmenorrhoea-related pain that occurs as a result of the intensity of myometrial contractions can be targeted at this site to reduce pain at these targets. A systematic review found that there is already a large population of women that use CBPMs to treat other gynaecological disorders—including chronic pelvic pain, vulvodynia, endometriosis, and gynaecology cancer-related pain [61]. This review found that most of the women from the studies reported improvement in pain scores. The results can be translated and applied to the management of dysmenorrhoea.

## 6. Clinical Trials on the Application of Cannabis-Based Medicines for Pain Relief

Table 2 summarises all clinical trials relating to CBPM and pain management. The search term “cannabis” AND “pain” was used in PubMed and results filtered for clinical trials; papers that were not relevant to pain were removed from the search. Of the 27 clinical trials described in Table 2, two-thirds of the trials reported a clear significant beneficial effect in the majority of their patients [62,63,64,65,66,67,68,69,70,71,72,73,74,75,76,77,78,79]. Four of the trials found no significant benefit in comparison with placebo [80,81,82,83]; therefore, overall owing to a clear majority significant benefit in the use of CBPM to treat pain.

Five of the trials had unclear outcomes. Of these, one trial found no effect on pain but significant effect on quality of sleep [84]. Another trial reported only 5 out of 16 patients experiencing significant analgesic effect [85]. One trial of patients with advanced cancer found only low dose CBPM, in comparison to high dose, having a significant analgesic effect [86]. The two-phase trial found a significant difference between treatment and placebo group at 10 weeks but no difference at 14 weeks [87]. Finally, one trial did not report in full and so it remains unclear if there was any significant effect on pain symptoms [88].

The trials included in the table have a vast spread of methodology in regard to patient number and follow-up endpoints, with the longest trial following patients for two years [71] and the shortest trial only investigating acute pain management and following patients for 48 h [80]. In addition, the type of pain investigated varied amongst the studies—including non-cancer chronic pain, acute pain—such as orthopaedic pain immediately following fractures, and cancer-related pains. A systematic review published in the BMJ assessed the application of cannabis for pain, concluding only a small improvement of pain with CBPM; however, the lack of distinguishing types of pain must be noted [89]. The outcomes of reviews such as this require careful consideration of the effect of cannabis on each type of pain specifically rather than grouping pain conditions together as a whole.

Previous systematic reviews note positive pain outcomes of CBPM, although further high-quality research to better investigate the effect on different types of pain would be beneficial [90]. A more recent meta-analysis published in the BMJ found very low certainty evidence for common adverse events of CBPM to manage pain, with few serious adverse events [91]. Overall, most recent systematic reviews have suggested low certainty and inconclusive evidence for the application of CBPM to manage chronic, non-cancer pain [92]. Further research in the field would support clinicians with a clear consensus on the efficacy of the drug.

**Table 2 ijms-23-16201-t002:** Extracted using search term “cannabis and pain” in PubMed and filtered for clinical trials, removed papers not relevant to pain. Number of patients included only those that completed the trial. DB: double blind; FU: follow-up; Pts, patients; SE: side effect; USA: United States of America.

Dose & Delivery Method	Brand	Country	No. of Pts	Type of Pain	Study Method	FU	Outcomes	SEs and Adverse Events	Year
Oral intake of 400 mg CBD		Australia	100	Acute, non-traumatic lower back pain	A randomised, DB, placebo-controlled clinical trial of 400 mg CBD.	48 h	No significant benefit of CBD to treat lower back pain.	Common SE: sedation, nausea and headache—seen in both treatment and placebo group.	2021 [80]
Sublingual THC-rich cannabis oil (24.44 mg/mL THC and 0.51 mg/mL CBD)		Brazil	17	Fibromyalgia pain	A randomised, DB, placebo-controlled clinical trial of cannabis oil, dose increased as per symptom relief.	8 weeks	Statistically significant reduction in pain symptoms following fibromyalgia questionnaire.	No AEs.	2020 [62]
Vaporised cannabis (4.4% THC and 4.9% CBD)		USA	23	Sickle cell disease	A randomised, DB, placebo-controlled clinical trial of vaporised cannabis. After one month of washout, the groups crossed over treatments.	5 days	No significant benefit with vaporised cannabis treatment.	Common SE: sedation, mostly mild and self-limiting.	2020 [81]
Oromucosal spray with 1:1 THC to CBD	Sativex^Ⓡ^	Italy	15	Multiple sclerosis pain	An open-labelled, uncontrolled clinical trial of cannabis spray in one group of voluntary pts.	6 weeks	Treatment with cannabinoids improved pain scores.	Three pts reported SE of mild drowsiness.	2020 [63]
Topical CBD cream with 250 mg/3 fl. Oz	Theramu Relieve CBD compound cream	USA	29	Peripheral neuropathy of lower extremities	A randomised, DB, placebo-controlled clinical trial of CBD cream applied up to four times daily.	4 weeks	Statistically significant reduction in pain intensity and unpleasantness symptoms in treatment group.	None recorded.	2020 [64]
Oral Hemp-derived soft gel tablets containing 15.7 mg CBD and 0.5 mg THC	Ananda Professional	USA	97	Chronic pain in relation to opiate dependence	A prospective, single-arm cohort study assessing effect of CBD-rich hemp soft gel tablets with two tablets (~30 mg CBD) daily.	8 weeks	Significant improvement in pain scores and reduction in opioid intake and reliance in the treatment group.	2 reports of drowsiness. 1 palpitations. 1 nausea. 1 heartburn. 1 nighttime anxiety and disturbed sleep.	2019 [65]
Inhalation: Bedrocan 22.4 mg THC: <1 mg CBD VS Bediol 13.4 mg THC: 17.8 mg CBD VS Bedrolite <1 mg THC: 18.4 mg CBD	Bedrocan International BV	The Netherlands	20	Fibromyalgia	A randomised, DB, placebo-controlled clinical trial with 4 groups. Three groups were given cannabis with varying ratios of THC and CBD.	10 weeks	Bediol, combination 1:1 THC: CBD, provided the greatest analgesic effect followed by Bedrocan.	Common SE: cough, sore throat, bad taste, dyspnoea, dizziness, nausea, and excess sleeping.	2019 [66]
CBD, route of delivery not recorded		Uruguay	7	Chronic pain secondary to kidney transplant	An uncontrolled, single-arm clinical study of CBD, initially 100 mg daily, increased in increments to maximum 300 mg per day.	3 weeks	Two pts had optimal response for pain management; four pts had partial response, and one pt recorded no change in pain symptoms.	Mild SE recorded which resolved with titration of dose. No AEs.	2018 [67]
Nabiximols oral mucosal spray THC 27 mg/mL and CBD 25 mg/mL	Sativex^®^	EU, UK, USA	291	Advanced cancer	A randomised, DB, placebo-controlled clinical trial of Nabiximols.	7 weeks	Overall no significant effect on pain except for pt group in USA, likely due to type of cancer.	Common SE: gastrointestinal symptoms (nausea and vomiting), and dizziness.	2018 [84]
Oral sublingual spray: THC 27 mg/mL and CBD 25 mg/mL	Sativex^®^	Italy	20	Multiple sclerosis	A double-arm clinical trial with two groups of pts with multiple sclerosis, one group experiencing chronic neuropathic pain versus pain free, both treated with cannabinoid spray.	4 weeks	After four weeks of treatment both pain scores and quality of life ratings improved dramatically.	Common SE: dizziness, dry mouth, nausea and weakness.	2016 [68]
Oral mucosal spray, one spray delivers 2.7 mg THC and 2.5 mg CBD		UK, Czech, Romania, Belgium, Canada	234	Neuropathic pain	An open-label extension clinical study, derived from two parent randomised controlled trials. All pts received cannabinoid spray in addition to their current analgesia.	42 weeks	Cannabinoid spray had a beneficial effect on treatment resistant pain, in addition pts did not seek to increase daily dose or take other drugs.	Common SE: dizziness and nausea.	2015 [69]
Oral mucosal spray, one spray delivers 2.7 mg THC and 2.5 mg CBD	Sativex^®^	UK, Czech Republic, Romania, Belgium, Canada	173	Peripheral neuropathic pain associated with allodynia	A randomised, DB, placebo-controlled clinical trial of cannabinoid sprays.	15 weeks	Statistically significant improvement in pain symptom scores for the pts in the treatment group.	Common SE: dizziness, nausea, fatigue and dysgeusia.	2014 [70]
Nabiximols oral mucosal spray THC 27 mg/mL and CBD 25 mg/mL	Sativex^®^	Canada	16	Chemotherapy induced neuropathic pain	A randomised, DB, placebo-controlled clinical trial of cannabinoid sprays.	6 months	No statistically significant in pain scores. 5 pts in the treatment group reported a clinically reduction in pain.	Common SE: dizziness, nausea, fatigue and dry mouth.	2014 [85]
Oral mucosal spray, one spray delivers 2.7 mg THC and 2.5 mg CBD	Sativex^®^	UK, Czech Republic, Canada, Spain, France	Phase A = 297 Phase B = 41	Central neuropathic pain secondary to multiple sclerosis	This was a two-phase study. Phase A was a randomised, DB, placebo-controlled clinical trial of cannabinoid spray. Phase B was an open plan treatment phase of cannabinoid spray.	33 weeks	Phase A saw a statistically significant improvement in pain scores in treatment group at week 10 but by week 14 there was no statistically significant difference between groups signifying placebo effect taking place. Phase B saw a statistically significant difference in pain scores with treatment.	Common SE: dizziness, fatigue, somnolence, vertigo and nausea.	2013 [87]
Oral mucosal spray, one spray delivers 2.7 mg THC and 2.5 mg CBD	Sativex^®^	UK, Belgium	43	Terminal cancer-related pain refractory to strong opioid analgesics	An open-label extension study derived from a parent randomised control trial.	2 years	At completion, 41 pts withdrew from the trial with one pt remained on treatment. The treatment group responded well long-term, did not reach tolerance and had no loss of effect over time.	Significant number of AEs but only one related to cannabis treatment. SE: dizziness, nausea, vomiting, somnolence and confusion.	2013 [71]
Oral mucosal spray, one spray delivers 2.7 mg THC and 2.5 mg CBD	Sativex^®^	Countries not specified, study across: North America, Europe, Latin America, South Africa	263	Opioid-treated cancer pts with poorly-controlled pain	A randomised, DB, placebo-controlled, graded-dose clinical trial divided into four arms. Three arms of the trials involved daily intake with varying doses of cannabinoid spray and the fourth arm treated with placebo.	9 weeks	No significant difference in the number of pts reaching primary endpoint, 20% reduction in pain scores. High dose treatment was not well tolerated with little or no analgesic effect. Low dose treatment achieved 26% improvement in secondary endpoint pain scores.	Common SE: nausea, dizziness, vomiting, somnolence and disorientations. AE: 12 pts suffered from neoplasm progression in the high-dose group compared to 24 in the low-dose group.	2012 [86]
Oral mucosal spray, one spray delivers 2.7 mg THC and 2.5 mg CBD VS oral mucosal spray with each spray delivering 2.7 mg THC	Sativex^®^	UK	144	Cancer-related pain	A randomised, DB, placebo-controlled clinical trial with three groups. One group of pts received THC: CBD extract, one group received THC extract and the final group received placebo.	2 weeks	THC: CBD spray was effective in managing pain not adequately relieved by opioids. No statistically significant difference in symptomatic pain between THC extract and placebo. No statistically significant outcome for secondary endpoints, sleep quality and reduced nausea.	Common SE: nausea, somnolence and dizziness.	2010 [72]
Oral mucosal spray, one spray delivers 2.7 mg THC and 2.5 mg CBD	Sativex^®^	UK	23	Diabetic neuropathy	A randomised, DB, placebo-controlled clinical trial of cannabinoid sprays.	12 weeks	No statistically significant improvement of pain scores in treatment group.	Six pts withdrew from the study due to AEs, not specified.	2009 [82]
Oral mucosal spray, one spray delivers 2.7 mg THC and 2.5 mg CBD	Sativex^®^	I	17	Secondary progressive multiple sclerosis	A randomised, DB, placebo-controlled crossover clinical trial. Pts were given either cannabinoid spray or placebo for three weeks, followed by a two-week washout period of no treatment, then three weeks on the opposite arm of the trial.	8 weeks	Neurophysiological findings suggest effective modulation of the nociceptive system with treatment, found to be concordant with pt pain scores, which also improved with treatment.	Common SE: drowsiness and slower thinking, dizziness and vertigo, nausea and vomiting, and fatigue.	2008 [73]
Oral mucosal spray, one spray delivers 2.7 mg THC and 2.5 mg CBD	Sativex^®^	UK	63	Multiple sclerosis	An uncontrolled, open-labelled five-week randomised clinical trial continued as an indefinite-duration extension. Pts allowed up to 48 sprays per day, titrated to this dose with a maximum 50% increase in doses each day.	(Indefinite) Mean duration recorded as 463 days	The cannabinoid spray was found to be effective in managing pain with no tolerance identified over long-term use.	Common SE: dizziness and nausea. Mild to moderate severity.	2007 [74]
Oral mucosal spray, one dose delivers 2.7 mg THC and 2.5 mg CBD	Sativex^®^	UK	125	Neuropathic pain	A randomised, DB, placebo-controlled clinical trial of oral mucosal cannabis sprays.	5 weeks	Statistically significant in favour of the cannabinoid spray. Pts in treatment group saw a 22% reduction in pain scores compared to 8% for placebo.	One or more mild SE in 91% of treatment group, mostly gastrointestinal, compared to 77% of placebo group.	2007 [75]
Oral mucosal spray, one spray delivers 2.7 mg THC and 2.5 mg CBD	Sativex^®^	UK	58	Rheumatoid arthritis	A randomised, DB, placebo-controlled clinical trial or oral mucosal cannabis sprays.	5 weeks	Statically significant analgesic activity in treatment group as well as significant reduction in disease activity.	The treatment group had no serious AE or SE. The placebo group had three withdrawals due to AE and two serious SE reported.	2006 [76]
Oral mucosal spray, one spray delivers 2.7 mg THC and 2.5 mg CBD	Sativex^®^	UK	64	Multiple sclerosis	A randomised, DB, placebo-controlled clinical trial of cannabinoid oral mucosal spray to manage central neuropathic pain secondary to multiple sclerosis.	5 weeks	Significantly improved pain scores and sleep disturbance in treatment group.	Common SE: dizziness, somnolence, and gastrointestinal symptoms. One AE of tachycardia, agitations and hypertension and one AE of paranoid ideation.	2005 [77]
Oral mucosal spray, one spray delivers 2.7 mg THC and 2.5 mg CBD VS oral mucosal spray with THC	Sativex^®^ and GW-2000-02	UK	141	Brachial plexus avulsion	A randomised, DB, placebo-controlled, single centre, three period crossover clinical trial. Three groups of pts—first group received THC: CBD; second group received greater ratio THC compared to CBD; and third group received placebo.	8 weeks	Statistically significant reduction in pain scores and improving sleep quality in both groups treated with cannabis based medicines compared to placebo.	Three withdrawals from the study. One for SE of nausea, second for SE of feeling faint and final due to anxiety and paranoia. No serious AE recorded.	2004 [43]
Oral mucosal spray, one spray delivers 2.7 mg THC and 2.5 mg CBD	Sativex^®^	UK	154	Multiple sclerosis	A randomised, DB, placebo-controlled clinical trial of cannabinoid sprays.	6 weeks	No statistically significant analgesic effect, likely due to a large placebo effect. However statistically significant improvement in spasticity symptoms recorded.	Common SE: dizziness, disturbance in attention, fatigue, disorientation, feeling drunk, somnolence and vertigo. No serious AE recorded.	2004 [83]
Oral mucosal spray, one spray delivers 2.7 mg THC and 2.5 mg CBD VS oral mucosal spray with CBD VS oral mucosal spray with THC		UK	34	Chronic pain	A randomised, DB, placebo-controlled clinical trial where groups were either treated with varying THC: CBD ratios of cannabinoid sprays or placebo.	12 weeks	Improvements in pain score seen for both THC and THC: CBD group. Unclear whether these were neuropathic pain improvement or secondary outcomes of improved mental health and quality of sleep.	Common SE: drowsiness and euphoria. No serious AE recorded.	2003 [88]
Oral cannabis extract VS THC extract		UK	611	Multiple sclerosis	A randomised, DB, placebo-controlled clinical trial of oral cannabis extract, including 33 centres in the UK.	15 weeks	Both the cannabis-based medicine and THC extract improved pain scores with no change in spasticity.	Common SE: dizziness, light-headedness and gastrointestinal symptoms.	2003 [78]

## 7. Cannabis-Based Medicines and Dysmenorrhoea

Currently, there is little evidence for the application of CBPMs to manage dysmenorrhoea. Using the search terms “cannabis AND dysmenorrhoea” in the PubMed, EMBASE, MEDLINE, Google Scholar, Cochrane Library (Wiley) databases and a clinical trial website (clinicaltrials.gov), there was only one paper published relevant to primary dysmenorrhoea and one ongoing trial. The paper investigates the perceptions and barriers of CBPM amongst women with primary dysmenorrhoea in Australia and finds that main concerns exist around stigma, driving rules and accessibility [93]. It can be derived that, due to their interaction with EC receptors, CBPMs may help manage menstrual pain as previously discussed [94]. In [93], social and online media have advertised the use of CBPM, specifically CBD due to being legal online, for the treatment of dysmenorrhoea. There are ongoing concerns about the effect of social media marketing on adolescents and their interaction with CBPM [95].

Currently, there are now several online start-ups that offer specifically CBD-related products alongside sanitary products to alleviate the pain experienced during the menstrual cycle. These CBD products are often offered in edible form, for example as chocolate. The brands, including Ohne^®^ and Beyou^®^, offer the CBD products without direct claims for the treatment of dysmenorrhoea; however, indirectly insinuating the advantages of CBD in the management of dysmenorrhoea. A lack of U.S. Food and Drug Administration (FDA) and Medicines and Healthcare products Regulatory Agency (MHRA) approval of CBD for dysmenorrhoea prevents the potentially controversial claims that may be considered misleading.

A novel product to enter the female focussed market includes a tampon coated with CBD oil, marketed by Daye^®^. There is currently no research published on the efficacy or safety of this product or any similar products. Due to the nature of the product as a tampon, not considered to be a medicine, no approval is required prior to marketing. This product has been designed to deliver the CBD directly to the pelvic region where pain signals are most active during menstruation. Although the design is extremely innovative, questions arise regarding the efficiency of drug delivery, as the tampon is required to both absorb menstrual blood effectively alongside the release of CBD to alleviate pain. The product would need to work alongside the theory of CBD, rather than CBPM, to manage dysmenorrhoea. The conventional tampon does create issues of the drug release properties of the tampon. A tampon aiming to release drugs and absorb blood will need to be redesigned with new materials that have such properties.

The mechanism of the product is called into question; the tampon sits in the vaginal canal, and it is unclear whether it is assumed to target the pain in the uterus through localisation or general absorption into the bloodstream. It is questionable whether the CBD reaches the uterus as a driver of the pain. In addition, once the tampon is inserted, the release of the drug is uncontrolled; in order to be used for pain relief, it is important for the delivery system to harness controlled and modified release of CBPM to manage pain over long periods. These questions will need to be addressed with appropriate clinical evidence in order for such products to access the wider market, worldwide.

There is currently only one clinical trial, as reported on ClinicalTrials.gov, assessing the application of CBD to manage dysmenorrhoea [96] and no other trials investigating the application of cannabis-based medicines for dysmenorrhoea. The trial is taking place in Michigan in the United States and is sponsored by Pure Green^®^, a medical cannabis company with a license to sell CBPM legally. It is currently in the recruitment phase and aiming to enlist 30 patients; there is one arm for the trial, so all subjects recruited will receive sublingual tablets containing 30 mg of CBD and 1 mg of THC. Although this trial is the first of its kind to investigate the potential use of this drug, there are flaws with the trial design. These include the short follow-up of only two months, lack of a control group to compare results, and groups with adjusted dosage. Response bias with self-reported pain may be resolved with placebo and control groups. Nevertheless, this remains a positive start for the future of CBPMs for the treatment of dysmenorrhoea.

## 8. Safety and Toxicology of Cannabis-Based Medicines

The safety of CBPM remains in question with a lack of assessment of the effect of long-term use and a suspicion for interference with brain development. Cannabis is a natural remedy, used for over 2500 years and worldwide by about 209 million people in the year 2022 as per the United Nations [97]. The mechanisms of CBPMs and their interaction with the EC system are not yet fully understood, and the diversity of CBPM products creates complexities as the effect of each differ slightly. Both short-term and long-term adverse events need to be assessed when investigating the safety of CBPMs.

The side effects of CBD and THC vary due to their distinct interactions with the EC system and its receptors. Short-term side effects of THC include dry mouth, dizziness, nausea, and somnolence, which dissipate as the drug is metabolised and have mostly been described as mild to moderate [98]. The short-term side effects of CBPM, although not acutely unsafe, may lead to dangerous behaviour, particularly when higher concentrations of THC are involved. This includes a higher risk of road traffic accidents and other high-risk activities effected by delayed psychomotor performance [99]. The effects of CBD include fatigue and somnolence, diarrhoea and low blood pressure, although, as with most drugs, these vary per individual [100]. Table 3 mentions the side effects of THC versus the CBD [101,102,103,104].

Cannabis has also been confirmed to have a role in drug-to-drug interactions. Cannabis-based products interact with the Cytochrome P450 (CP450) enzyme, with CBD and THC both being potent inhibitors of the CP450 enzyme; hence, doses of other medicinal substrates of this enzyme will need to be carefully increased, such as the commonly prescribed medications Warfarin and Diazepam [105].

There are public health concerns regarding the addictive potential of cannabis, particularly with greater concentrations of THC and its withdrawal potential [106]. There is little information on the lethal doses of CBPM in adults unless triggering myocardial infarction leading to death, and it is generally considered that a lethal overdose of CBPM is an unlikely event; note that monkeys can tolerate 9000 mg/kg THC without fatal consequences [37]. However, there are concerns regarding neurotoxicity in paediatric cases of acute toxicity with once case of a fatal outcome [107]. The effects of THC on mood have been considered to increase the risk of suicide; however, a meta-analysis found that acute toxicity does not increase the risk of suicide and a lack of homogeneity in study results investigating the relationship between cannabis use and suicide outcomes [108]. It is important to note that the inhalation method of cannabis delivery may carry increased risks of oral disease, with research indicating that smoking cannabis risks cancer of oral mucosa, dental caries, periodontal disease and oral infections [109]. The effects of cannabis use on lung disease are difficult to analyse due to variability in method of inhalation and concurrent tobacco use; however, evidence does show that cannabis smoke leads to bronchitis and bullous lung disease with a greater risk of pneumothorax [110].

## 9. Barriers to Use of Cannabis-Based Medicines

There are several barriers preventing further clinical application of CBPMs and its growth in healthcare provision. As per the evidence presented above, there remains a lack of consensus for the application of CBPMs and clinical efficacy of cannabis-based products—this is the first and foremost major barrier to proper implementation of CBPMs in clinical practice. Currently, the NHS in the UK does not provide CBPMs to patients suffering from long-term pain, as the evidence is not strong enough. For the NHS to implement CBPM for pain, a clear consensus is required with a cost–risk analysis.

The barriers are multifaceted and include stigmatic attitudes towards cannabis as a drug; current high cost of prescription drugs; fear of psychosis and developmental delay; lack of education for doctors on the indications and process of prescribing CBPM; and restrictions with guidelines defining CBPM as a “last resort” treatment—in the UK [8]. Note that each country is experiencing different situations regarding the legislation and clinical practice of CBPM use.

There has been a plateau in the number of prescriptions in the UK since legalisation in 2018 evident from an independent NHS report [111]. It is unclear whether this is secondary to safety concerns on behalf of doctors or lack of awareness of the change in law and scheduling of the drug. Further education for healthcare professionals is required to prevent misconceptions and stigmatisation, with aim of improving doctors’ attitudes [9,10]. Some advice is for laws to continue to change to provide clear guidelines on how and when to prescribe CBPM and to reduce the number of gatekeeping steps required [112].

Across many countries, private prescriptions of CBPM are a high-cost burden that many patients will not be able to afford. Insurance companies and national health services often do not recognise the need for CBPMs and will not pay for these costs. The cost of prescriptions is high as CBPMs may often be imported with additional taxes, and for many patients this may be a lifelong cost to pay [113].

There has been a rise in the OTC sales of CBD products in recent years, with multiple countries relaxing policies surrounding CBPM or novelty OTC CBD products [5]. OTC products only allow negligible concentrations of the CBD drug; therefore questioning if there is any meaningful health impact, as larger quantities remain illegal for tender unless with medical prescription. There are several companies selling various CBD products for food or cosmetic benefit. These include oils, balms, creams, mouth sprays, patches and infused bottled water, coffee or soda drinks. The safety of these products is questionable as they are sold as “novel food”, not advertised for health benefits, and so lack rigorous safety testing.

## 10. Conclusions

Cannabis-based products are entering both healthcare and commercial markets with a surge in availability. Multiple companies are now marketing cannabidiol products alongside sanitary products for potential analgesic effects on dysmenorrhoea. This leads to considering the application of cannabis-based medicines in managing dysmenorrhoea.

For the successful application of cannabis-based medicines for dysmenorrhoea, a better understanding of its mechanisms of action and safety profile is required. Further research should investigate appropriate dosage, titration method and route of administration—in order to establish a protocol for treatment. Clinicians may struggle to prescribe cannabis-based medicines with a lack of education and stigmatisation amongst healthcare staff; therefore, providing a clear set of guidelines would support its implementation. Prior to integration in routine clinical practice, a consensus on the applications of cannabis-based medicines and comprehensive safety profile is required. With further research, cannabis-based medicines may become the norm in the management of severe or treatment-resistant dysmenorrhoea.

## Figures and Tables

**Figure 1 ijms-23-16201-f001:**
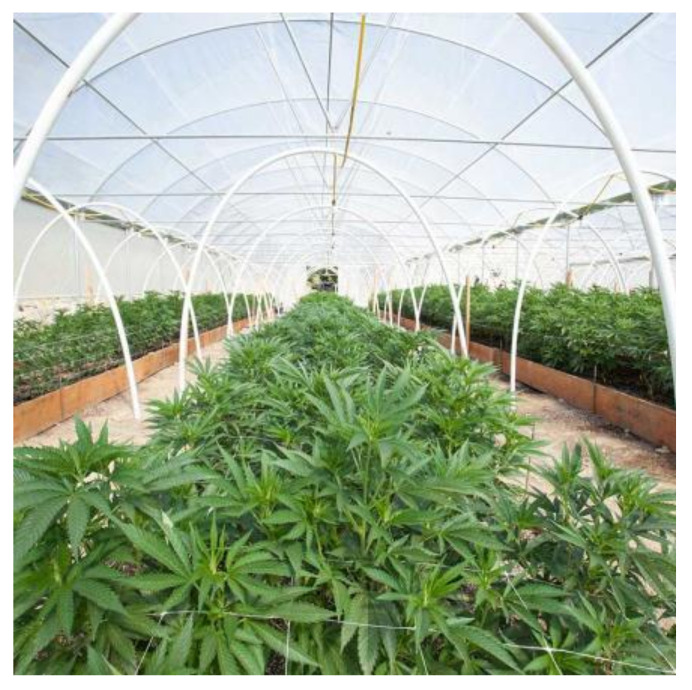
Cannabis plants being grown and cultivated in a legal cannabis farm. *Image Credit: Canna Obscura/Shutterstock.com* (accessed on 20 October 2022).

**Figure 2 ijms-23-16201-f002:**
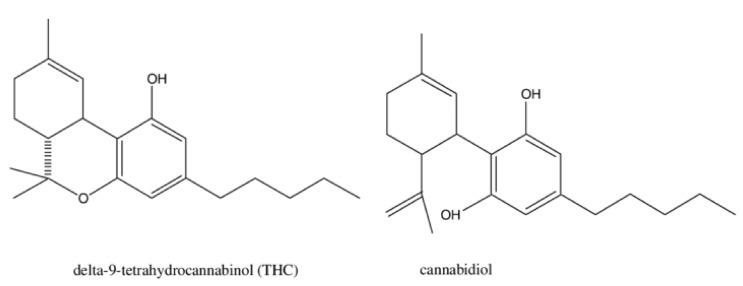
Chemical structure of THC and CBD.

**Table 1 ijms-23-16201-t001:** Summary of differences between oral versus inhaled administration of CBPMs.

Method of Absorption of Cannabinoids	Oral Ingestion	Inhalation
**Mechanism of absorption of cannabinoids**	Via gastrointestinal system followed by liver “first pass” metabolism	Into the bloodstream from lung periphery, following inhalation
**Time to peak concentration of cannabinoids**	120 min	3–10 min
**Duration of action of cannabinoids**	4–12 h	2–3 h
**Bioavailability of cannabinoids**	6%	10–35%

**Table 3 ijms-23-16201-t003:** Comparing the common short-term side effects of THC versus CBD.

Side Effects	
THC	CBD
Somnolence	Somnolence
Nausea	Nausea
Dizziness and vertigo	Dizziness
Loss of concentration	Loss of concentration
Dry mouth	Dry mouth
Amnesia	Fatigue
Anxiety	Diarrhoea with abdominal pain

## Data Availability

Not applicable.

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
