# Peer review of "Dysmenorrhoea: Can Medicinal Cannabis Bring New Hope for a Collective Group of Women Suffering in Pain, Globally?"

_ijms, 2022, doi:10.3390/ijms232416201_

Round 1

Reviewer 1 Report

This is a paper on cannabis-based products and dysmenorrhoea. While a great deal of work has been produced, I do not see what the article add to the litterature.

As systematic reviews and meta-analyses have already been published on CBP and chronic pain, to me there is no need to re-review the litterature (especially if the article is not presented as a systematic review), especially on a less robust way than wat has been done.

The special focus that is done on the UK is not justified.

Lengthy generalist parts on cannabis and CBP extraction does not seem appropriate.

While the manuscript does not fulfill criteria for publication to me as it is, I acknowledge that the topic of CBP for dysmenorrhoea deserves an article if nothing has been published. However, the manuscript would need to be re-written and give larger room to dysmenorrhoea (mechanisms, comorbid symptoms, relationships with the endocannabinoid system, proper systematic review...).

Please find many more minor comments in the attached file.

Author Response

This is a paper on cannabis-based products and dysmenorrhea. While a great deal of work has been produced. I do not see what the article add to the litterature.

As systematic reviews and meta-analyses have already been published on CBP and chronic pain, to me there is no need to re-review the litterature (especially if the article is not presented as a systematic review), especially on a less robust way than wat has been done.

The special focus that is done on the UK is not justified.

Response: We would like to thank the reviewer for the considerable amount of time and effort spent on our manuscript. We have responded to all of the comments and revised the manuscript accordingly and this has impacted the clarity and significantly improved the manuscript.

In response to comments regarding special focus on the UK - as this is an emerging technology and new field we believe if would be useful to comment on legislation and clinical practice. We are a UK based research team and gynaecologists and so have expertise in the practice of the National Health Service in the UK. We have reduced, where possible, the focus on the UK but we believe sharing the scope of current practice would give reader’s a better understanding of the current situation. Unfortunately, we do not feel we can closely comment on the situation in other countries specifically on the application of cannabis for pelvic pain as we have not experienced it and there is limited research available due to the novel application.

Lengthy generalist parts on cannabis and CBP extraction does not seem appropriate.

Response: We have revised this section by reducing the amount of information provided. We believe a summary of CBP extraction is essential to help the reader understand the process and hence the significance of purity and difficulties of quality control.

While the manuscript does not fulfill criteria for publication to me as it is, I acknowledge that the topic of CBP for dysmenorrhea deserves an article if nothing has been published.

Response: We would like to thank the reviewer for acknowledging the importance of CBP in treating the unmet clinical need of dysmenorrhea.

However, the manuscript would need to be re-written and give larger room to dysmenorrhea (mechanisms, comorbid symptoms, relationships with the endocannabinoid system, proper systematic review...).

Response: We have revised the manuscript according to the comments. We acknowledge the condition of dysmenorrhoea as an unmet clinical need with CBP offering a potential solution. Up to 90% of women of reproductive age suffer from dysmenorrhoea worldwide leading to large impacts mental and sexual health and work absenteeism.

The majority of women suffer in silence due to the taboo nature of the subject of menstruation. Focus of research has historically been on Men’s Health, with Women’s Health falling behind as a result causing a gender health gap. But in recent years there has been a shift to prioritise Women’s Health innovation leading to significant advancements in the field.

There is an insufficient number of original papers to carry out a systemic review of CBP for dysmenorrhoea. CBP for dysmenorrhoea is considered an emerging strategy with less than 10 relevant publications on the topic. We have founded a company, see Liberum Health (www.liberumhealth.com), to overcome problems with the gender health gap and to improve the management of dysmenorrhoea.

Please find many more minor comments in the attached file.

Response: We would like to thank the reviewer again for the significant amount of time and effort spent on our manuscript. We have revised the manuscript according to the minor comments with track changes.

Reviewer 2 Report

The article is well written and well organized. It covers key aspects in relation to the management of dysmenorrhea with cannabis-based approaches to modulate and reduce pain. I'd suggest the authors to consider the following minor revisions:

1. Rephrase sentence in lines 32-33. As it is, is not clear.

2. Specify the abbreviation of NSAIDS at first use. 

3. Rephrase or expand sentence in lines 61-62. "yet little is understood about this complex drugs" can be confusing, as the huge number of evidence on the Endocannabinoid System and cannabis are proof of the progress in understanding it's mechanism of action in physiological and pathological conditions. 

4. I'd suggest the authors to expand Paragraph 4. It'd be useful to the reader to find here more reference and en expanded description of the effects of cannabis/cannabinoids in the management of pain.

Author Response

The article is well written and well organized. It covers key aspects in relation to the management of dysmenorrhea with cannabis-based approaches to modulate and reduce pain. I'd suggest the authors to consider the following minor revisions:

Response: We would like to thank the reviewer for their positive comments.

  1. Rephrase sentence in lines 32-33. As it is, is not clear.

Response: The sentence in lines 32-33 has been rephrased, please see the following revised sentences.

Primary dysmenorrhoea occurs as a result of increased prostaglandin release, amongst other chemicals, causing abnormal uterine contractions, interrupting blood flow and increasing anaerobic metabolites that in turn, stimulate pain receptors (2). Secondary dysmenorrhoea occurs as a result of underlying gynaecological disease, such as endometriosis.

  1. Specify the abbreviation of NSAIDS at first use.

Response: NSAIDS has been expanded to non-steroidal anti-inflammatory drugs.

  1. Rephrase or expand sentence in lines 61-62. "yet little is understood about this complex drugs" can be confusing, as the huge number of evidence on the Endocannabinoid System and cannabis are proof of the progress in understanding it's mechanism of action in physiological and pathological conditions.

Response: We have rephrased the sentence according to the comments.

  1. I'd suggest the authors to expand Paragraph 4. It'd be useful to the reader to find here more reference and en expanded description of the effects of cannabis/cannabinoids in the management of pain.

Response: We would like to thank the reviewer for their comment. We have expanded the paragraph to include the effects of cannabis on managing pain in reference to section 4 which provides further detail on mechanisms.

Round 2

Reviewer 1 Report

I thank the authors for the subtantial improvements they made.

Please find further comments in the attached document.

It would be great to give dysmenorrhea/pain a more central place within the manuscript.

Author Response

Response: We would like to thank the reviewer for the time and effort spent on the manuscript. We have made new revisions to further update the manuscript and believe it has improved again greatly. The majority of comments have been resolved with revisions as attached in latest version of the manuscript. Below are responses to a few comments where changes could not be made.

Line 59: Note has been made on other analgesic/holistic therapies however we believe to cover these in deserved depth, a separate paper would be most appropriate focussing on all options available and stage of use in clinical practice.

Line 81: In regard to the motivations for the cannabis prescriptions in the UK, there is no information on this at present. Likelihood is that prescription is secondary to epilepsy, N&V or pain however we felt it best not to present such assumptions in the paper.

Line 165: Note focus of this paper is to THC and CBD so other phytochemicals not mentioned.

Line 170: We are unable to find specific data on % extract for the different methods however understandably there are a number of variable factors which would affect the results.

Line 274: Endometriosis is not further discussed as the focus of this paper is on primary dysmenorrhoea. Although we have now highlighted use of CBPM and gynaecological conditions in general.

Line 276: Unfortunately, we are unable to source any specific data regarding CBD sales online due to the extent of reach.

Round 3

Reviewer 1 Report

I thanks the authors for those improvements.

I only have minor comments, in the attached file.

Author Response

We thank the reviewer kindly for their comments and for all their help thus far in improving the quality of the paper. We have revised the paper according to the reviewers comments.

In regard to line 229, developing the description of the role of CBPM on the myometrium is limited as there is very little research on the subject matter, although we have tried to include information available.

Thanks again to the reviewer for their contribution to improving the paper.